# Deep Recursive Neural Networks
# for Compositionality in Language

**Ozan İrsoy**
Department of Computer Science
Cornell University
Ithaca, NY 14853
`oirsoy@cs.cornell.edu`

**Claire Cardie**
Department of Computer Science
Cornell University
Ithaca, NY 14853
`cardie@cs.cornell.edu`

## Abstract

Recursive neural networks comprise a class of architecture that can operate on structured input. They have been previously successfully applied to model compositionality in natural language using parse-tree-based structural representations. Even though these architectures are deep in structure, they lack the capacity for hierarchical representation that exists in conventional deep feed-forward networks as well as in recently investigated deep recurrent neural networks. In this work we introduce a new architecture — a *deep recursive neural network* (deep RNN) — constructed by stacking multiple recursive layers. We evaluate the proposed model on the task of fine-grained sentiment classification. Our results show that deep RNNs outperform associated shallow counterparts that employ the same number of parameters. Furthermore, our approach outperforms previous baselines on the sentiment analysis task, including a multiplicative RNN variant as well as the recently introduced paragraph vectors, achieving new state-of-the-art results. We provide exploratory analyses of the effect of multiple layers and show that they capture different aspects of compositionality in language.

## 1    Introduction

Deep connectionist architectures involve many layers of nonlinear information processing [1]. This allows them to incorporate meaning representations such that each succeeding layer potentially has a more abstract meaning. Recent advancements in efficiently training deep neural networks enabled their application to many problems, including those in natural language processing (NLP). A key advance for application to NLP tasks was the invention of *word embeddings* that represent a single word as a dense, low-dimensional vector in a meaning space [2], and from which numerous problems have benefited [3, 4].

**Recursive neural networks**, comprise a class of architecture that operates on structured inputs, and in particular, on directed acyclic graphs. A recursive neural network can be seen as a generalization of the recurrent neural network [5], which has a specific type of skewed tree structure (see Figure 1). They have been applied to parsing [6], sentence-level sentiment analysis [7, 8], and paraphrase detection [9]. Given the structural representation of a sentence, e.g. a parse tree, they recursively generate parent representations in a bottom-up fashion, by combining tokens to produce representations for phrases, eventually producing the whole sentence. The sentence-level representation (or, alternatively, its phrases) can then be used to make a final classification for a given input sentence — e.g. whether it conveys a positive or a negative sentiment.

Similar to how recurrent neural networks are deep in time, recursive neural networks are deep in structure, because of the repeated application of recursive connections. Recently, the notions of *depth in time* — the result of recurrent connections, and *depth in space* — the result of stacking

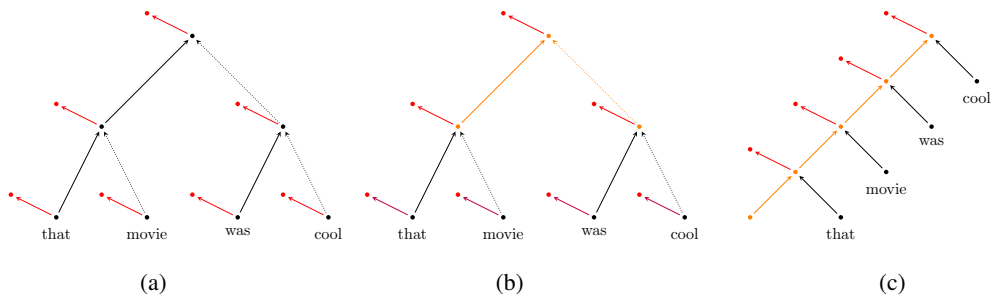

Figure 1: Operation of a recursive net (a), untied recursive net (b) and a recurrent net (c) on an example sentence. Black, orange and red dots represent input, hidden and output layers, respectively. Directed edges having the same color-style combination denote shared connections.

multiple layers on top of one another, are distinguished for recurrent neural networks. In order to combine these concepts, *deep* recurrent networks were proposed [10, 11, 12]. They are constructed by stacking multiple recurrent layers on top of each other, which allows this extra notion of depth to be incorporated into temporal processing. Empirical investigations showed that this results in a natural hierarchy for how the information is processed [12]. Inspired by these recent developments, we make a similar distinction between *depth in structure* and *depth in space*, and to combine these concepts, propose the *deep recursive neural network*, which is constructed by stacking multiple recursive layers.

The architecture we study in this work is essentially a deep feedforward neural network with an additional structural processing within each layer (see Figure 2). During forward propagation, information travels through the structure within each layer (because of the recursive nature of the network, weights regarding structural processing are shared). In addition, every node in the structure (i.e. in the parse tree) feeds its own hidden state to its counterpart in the next layer. This can be seen as a combination of feedforward and recursive nets. In a shallow recursive neural network, a single layer is responsible for learning a representation of composition that is both useful and sufficient for the final decision. In a deep recursive neural network, a layer can learn some parts of the composition to apply, and pass this intermediate representation to the next layer for further processing for the remaining parts of the overall composition.

To evaluate the performance of the architecture and make exploratory analyses, we apply deep recursive neural networks to the task of fine-grained sentiment detection on the recently published Stanford Sentiment Treebank (SST) [8]. SST includes a supervised sentiment label for every node in the binary parse tree, not just at the root (sentence) level. This is especially important for deep learning, since it allows a richer supervised error signal to be backpropagated across the network, potentially alleviating vanishing gradients associated with deep neural networks [13].

We show that our deep recursive neural networks outperform shallow recursive nets of the same size in the fine-grained sentiment prediction task on the Stanford Sentiment Treebank. Furthermore, our models outperform multiplicative recursive neural network variants, achieving new state-of-the-art performance on the task. We conduct qualitative experiments that suggest that each layer handles a different aspect of compositionality, and representations at each layer capture different notions of similarity.

## 2 Methodology

### 2.1 Recursive Neural Networks

Recursive neural networks (e.g. [6]) (RNNs) comprise an architecture in which the same set of weights is recursively applied within a structural setting: given a positional directed acyclic graph, it visits the nodes in topological order, and recursively applies transformations to generate further representations from previously computed representations of children. In fact, a recurrent neural network is simply a recursive neural network with a particular structure (see Figure 1c). Even though

RNNs can be applied to any positional directed acyclic graph, we limit our attention to RNNs over positional binary trees, as in [6].

Given a binary tree structure with leaves having the initial representations, e.g. a parse tree with word vector representations at the leaves, a recursive neural network computes the representations at each internal node $\eta$ as follows (see also Figure 1a):

$$x_\eta = f(W_L x_{l(\eta)} + W_R x_{r(\eta)} + b) \tag{1}$$

where $l(\eta)$ and $r(\eta)$ are the left and right children of $\eta$, $W_L$ and $W_R$ are the weight matrices that connect the left and right children to the parent, and $b$ is a bias vector. Given that $W_L$ and $W_R$ are square matrices, and not distinguishing whether $l(\eta)$ and $r(\eta)$ are leaf or internal nodes, this definition has an interesting interpretation: initial representations at the leaves and intermediate representations at the nonterminals lie in the same space. In the parse tree example, a recursive neural network combines the representations of two subphrases to generate a representation for the larger phrase, in the same meaning space [6]. We then have a task-specific output layer above the representation layer:

$$y_\eta = g(U x_\eta + c) \tag{2}$$

where $U$ is the output weight matrix and $c$ is the bias vector to the output layer. In a supervised task, $y_\eta$ is simply the prediction (class label or response value) for the node $\eta$, and supervision occurs at this layer. As an example, for the task of sentiment classification, $y_\eta$ is the predicted sentiment label of the phrase given by the subtree rooted at $\eta$. Thus, during supervised learning, initial external errors are incurred on $y$, and backpropagated from the root, toward leaves [14].

## 2.2 Untying Leaves and Internals

Even though the aforementioned definition, which treats the leaf nodes and internal nodes the same, has some attractive properties (such as mapping individual words and larger phrases into the same meaning space), in this work we use an untied variant that distinguishes between a leaf and an internal node. We do this by a simple parametrization of the weights $W$ with respect to whether the incoming edge emanates from a leaf or an internal node (see Figure 1b in contrast to 1a, color of the edges emanating from leaves and internal nodes are different):

$$h_\eta = f(W_L^{l(\eta)} h_{l(\eta)} + W_R^{r(\eta)} h_{r(\eta)} + b) \tag{3}$$

where $h_\eta = x_\eta \in \mathcal{X}$ if $\eta$ is a leaf and $h_\eta \in \mathcal{H}$ otherwise, and $W_\cdot^\eta = W_\cdot^{xh}$ if $\eta$ is a leaf and $W_\cdot^\eta = W_\cdot^{hh}$ otherwise. $\mathcal{X}$ and $\mathcal{H}$ are vector spaces of words and phrases, respectively. The weights $W_\cdot^{xh}$ act as a transformation from word space to phrase space, and $W_\cdot^{hh}$ as a transformation from phrase space to itself.

With this untying, a recursive network becomes a generalization of the Elman type recurrent neural network with $h$ being analogous to the hidden layer of the recurrent network (memory) and $x$ being analogous to the input layer (see Figure 1c). Benefits of this untying are twofold: (1) Now the weight matrices $W_\cdot^{xh}$, and $W_\cdot^{hh}$ are of size $|h| \times |x|$ and $|h| \times |h|$ which means that we can use large pretrained word vectors and a small number of hidden units without a quadratic dependence on the word vector dimensionality $|x|$. Therefore, small but powerful models can be trained by using pretrained word vectors with a large dimensionality. (2) Since words and phrases are represented in different spaces, we can use rectifier activation units for $f$, which have previously been shown to yield good results when training deep neural networks [15]. Word vectors are dense and generally have positive and negative entries whereas rectifier activation causes the resulting intermediate vectors to be sparse and nonnegative. Thus, when leaves and internals are represented in the same space, a discrepancy arises, and the same weight matrix is applied to both leaves and internal nodes and is expected to handle both sparse and dense cases, which might be difficult. Therefore separating leaves and internal nodes allows the use of rectifiers in a more natural manner.

## 2.3 Deep Recursive Neural Networks

Recursive neural networks are deep in structure: with the recursive application of the nonlinear information processing they become as deep as the depth of the tree (or in general, DAG). However, this notion of depth is unlikely to involve a hierarchical interpretation of the data. By applying

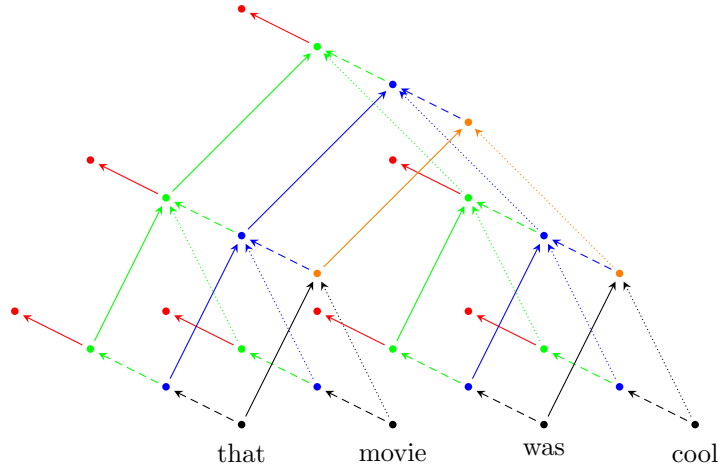

that     movie     was     cool

Figure 2: Operation of a 3-layer **deep recursive neural network**. Red and black points denote output and input vectors, respectively; other colors denote intermediate memory representations. Connections denoted by the same color-style combination are shared (i.e. share the same set of weights).

the same computation recursively to compute the contribution of children to their parents, and the same computation to produce an output response, we are, in fact, representing every internal node (phrase) in the same space [6, 8]. However, in the more conventional *stacked* deep learners (e.g. deep feedforward nets), an important benefit of depth is the hierarchy among hidden representations: every hidden layer conceptually lies in a different representation space and potentially is a more abstract representation of the input than the previous layer [1].

To address these observations, we propose the *deep* recursive neural network, which is constructed by stacking multiple layers of individual recursive nets:

$$h_\eta^{(i)} = f(W_L^{(i)} h_{l(\eta)}^{(i)} + W_R^{(i)} h_{r(\eta)}^{(i)} + V^{(i)} h_\eta^{(i-1)} + b^{(i)}) \qquad (4)$$

where $i$ indexes the multiple stacked layers, $W_L^{(i)}$, $W_R^{(i)}$, and $b^{(i)}$ are defined as before within each layer $i$, and $V^{(i)}$ is the weight matrix that connects the $(i-1)$th hidden layer to the $i$th hidden layer.

Note that the untying that we described in Section 2.2 is only necessary for the first layer, since we can map both $x \in \mathcal{X}$ and $h^{(1)} \in \mathcal{H}^{(1)}$ in the first layer to $h^{(2)} \in \mathcal{H}^{(2)}$ in the second layer using separate $V^{(2)}$ for leaves and internals ($V^{xh(2)}$ and $V^{hh(2)}$). Therefore every node is represented in the same space at layers above the first, regardless of their "leafness". Figure 2 provides a visualization of weights that are untied or shared.

For prediction, we connect the output layer to only the final hidden layer:

$$y_\eta = g(U h_\eta^{(\ell)} + c) \qquad (5)$$

where $\ell$ is the total number of layers. Intuitively, connecting the output layer to only the last hidden layer forces the network to represent enough high level information at the final layer to support the supervised decision. Connecting the output layer to all hidden layers is another option; however, in that case multiple hidden layers can have synergistic effects on the output and make it more difficult to qualitatively analyze each layer.

Learning a deep RNN can be conceptualized as interleaved applications of the conventional back-propagation across multiple layers, and backpropagation through structure within a single layer. During backpropagation a node $\eta$ receives error terms from both its parent (through structure), and from its counterpart in the higher layer (through space). Then it further backpropagates that error signal to both of its children, as well as to its counterpart in the lower layer.

# 3  Experiments

## 3.1  Setting

**Data.**  For experimental evaluation of our models, we use the recently published Stanford Sentiment Treebank (SST) [8], which includes labels for 215,154 phrases in the parse trees of 11,855 sentences, with an average sentence length of 19.1 tokens. Real-valued sentiment labels are converted to an integer ordinal label in $\{0, \ldots, 4\}$ by simple thresholding. Therefore the supervised task is posed as a 5-class classification problem. We use the single training-validation-test set partitioning provided by the authors.

**Baselines.**  In addition to experimenting among deep RNNs of varying width and depth, we compare our models to previous work on the same data. We use baselines from [8]: a naive bayes classifier that operates on bigram counts (BINB), shallow RNN (RNN) [6, 7] that learns the word vectors from the supervised data and uses $\tanh$ units, in contrast to our shallow RNNs, a matrix-vector RNN in which every word is assigned a matrix-vector pair instead of a vector, and composition is defined with matrix-vector multiplications (MV-RNN) [16], and the multiplicative recursive net (or the recursive neural tensor network) in which the composition is defined as a bilinear tensor product (RNTN) [8]. Additionally, we use a method that is capable of generating representations for larger pieces of text (PARAGRAPH VECTORS) [17], and the dynamic convolutional neural network (DCNN) [18]. We use the previously published results for comparison using the same training-development-test partitioning of the data.

**Activation Units.**  For the output layer, we employ the standard softmax activation: $g(x) = e^{x_i} / \sum_j e^{x_j}$. For the hidden layers we use the rectifier linear activation: $f(x) = \max\{0, x\}$. Experimentally, rectifier activation gives better performance, faster convergence, and sparse representations. Previous work with rectifier units reported good results when training deep neural networks, with no pre-training step [15].

**Word Vectors.**  In all of our experiments, we keep the word vectors fixed and do not finetune for simplicity of our models. We use the publicly available 300 dimensional word vectors by [19], trained on part of the Google News dataset ($\sim$100B words).

**Regularizer.**  For regularization of the networks, we use the recently proposed dropout technique, in which we randomly set entries of hidden representations to 0, with a probability called the dropout rate [20]. Dropout rate is tuned over the development set out of $\{0, 0.1, 0.3, 0.5\}$. Dropout prevents learned features from co-adapting, and it has been reported to yield good results when training deep neural networks [21, 22]. Note that dropped units are shared: for a single sentence and a layer, we drop the same units of the hidden layer at each node.

Since we are using a non-saturating activation function, intermediate representations are not bounded from above, hence, they can explode even with a strong regularization over the connections, which is confirmed by preliminary experiments. Therefore, for stability reasons, we use a small fixed additional L2 penalty ($10^{-5}$) over both the connection weights and the unit activations, which resolves the explosion problem.

**Network Training.**  We use stochastic gradient descent with a fixed learning rate (.01). We use a diagonal variant of AdaGrad for parameter updates [23]. AdaGrad yields a smooth and fast convergence. Furthermore, it can be seen as a natural tuning of individual learning rates per each parameter. This is beneficial for our case since different layers have gradients at different scales because of the scale of non-saturating activations at each layer (grows bigger at higher layers). We update weights after minibatches of 20 sentences. We run 200 epochs for training. Recursive weights within a layer ($W^{hh}$) are initialized as $0.5I + \epsilon$ where $I$ is the identity matrix and $\epsilon$ is a small uniformly random noise. This means that initially, the representation of each node is approximately the mean of its two children. All other weights are initialized as $\epsilon$. We experiment with networks of various sizes, however we have the same number of hidden units across multiple layers of a single RNN. When we increase the depth, we keep the overall number of parameters constant, therefore deeper networks become narrower. We do not employ a pre-training step; deep architectures are trained with the supervised error signal, even when the output layer is connected to only the final hidden layer.

| $\ell$ | $\lvert h\rvert$ | Fine-grained | Binary |
|---|---|---|---|
| 1 | 50 | 46.1 | 85.3 |
| 2 | 45 | 48.0 | 85.5 |
| 3 | 40 | 43.1 | 83.5 |
| 1 | 340 | 48.1 | 86.4 |
| 2 | 242 | 48.3 | 86.4 |
| 3 | 200 | 49.5 | **86.7** |
| 4 | 174 | **49.8** | 86.6 |
| 5 | 157 | 49.0 | 85.5 |

(a) Results for RNNs. $\ell$ and $\lvert h\rvert$ denote the depth and width of the networks, respectively.

| Method | Fine-grained | Binary |
|---|---|---|
| Bigram NB | 41.9 | 83.1 |
| RNN | 43.2 | 82.4 |
| MV-RNN | 44.4 | 82.9 |
| RNTN | 45.7 | 85.4 |
| DCNN | 48.5 | 86.8 |
| Paragraph Vectors | 48.7 | **87.8** |
| DRNN (4, 174) | **49.8** | 86.6 |

(b) Results for previous work and our best model (DRNN).

Table 1: Accuracies for 5-class predictions over SST, at the sentence level.

Additionally, we employ early stopping: out of all iterations, the model with the best development set performance is picked as the final model to be evaluated.

## 3.2 Results

**Quantitative Evaluation.** We evaluate on both fine-grained sentiment score prediction (5-class classification) and binary (positive-negative) classification. For binary classification, we do not train a separate network, we use the network trained for fine-grained prediction, and then decode the 5 dimensional posterior probability vector into a binary decision which also effectively discards the neutral cases from the test set. This approach solves a harder problem. Therefore there might be room for improvement on binary results by separately training a binary classifier.

Experimental results of our models and previous work are given in Table 1. Table 1a shows our models with varying depth and width (while keeping the overall number of parameters constant within each group). $\ell$ denotes the depth and $\lvert h\rvert$ denotes the width of the networks (i.e. number of hidden units in a single hidden layer).

We observe that shallow RNNs get an improvement just by using pretrained word vectors, rectifiers, and dropout, compared to previous work (48.1 vs. 43.2 for the fine-grained task, see our shallow RNN with $\lvert h\rvert = 340$ in Table 1a and the RNN from [8] in Table 1b). This suggests a validation for untying leaves and internal nodes in the RNN as described in Section 2.2 and using pre-trained word vectors.

Results on RNNs of various depths and sizes show that deep RNNs outperform single layer RNNs with approximately the same number of parameters, which quantitatively validates the benefits of deep networks over shallow ones (see Table 1a). We see a consistent improvement as we use deeper and narrower networks until a certain depth. The 2-layer RNN for the smaller networks and 4-layer RNN for the larger networks give the best performance with respect to the fine-grained score. Increasing the depth further starts to cause a degrade. An explanation for this might be the decrease in width dominating the gains from an increased depth.

Furthermore, our best deep RNN outperforms previous work on both the fine-grained and binary prediction tasks, and outperforms Paragraph Vectors on the fine-grained score, achieving a new state-of-the-art (see Table 1b).

We attribute an important contribution of the improvement to dropouts. In a preliminary experiment with simple L2 regularization, a 3-layer RNN with 200 hidden units each achieved a fine-grained score of 46.06 (not shown here), compared to our current score of 49.5 with the dropout regularizer.

**Input Perturbation.** In order to assess the scale at which different layers operate, we investigate the response of all layers to a perturbation in the input. A way of perturbing the input might be an addition of some noise, however with a large amount of noise, it is possible that the resulting noisy input vector is outside of the manifold of meaningful word vectors. Therefore, instead, we simply pick a word from the sentence that carries positive sentiment, and alter it to a set of words that have sentiment values shifting towards the negative direction.

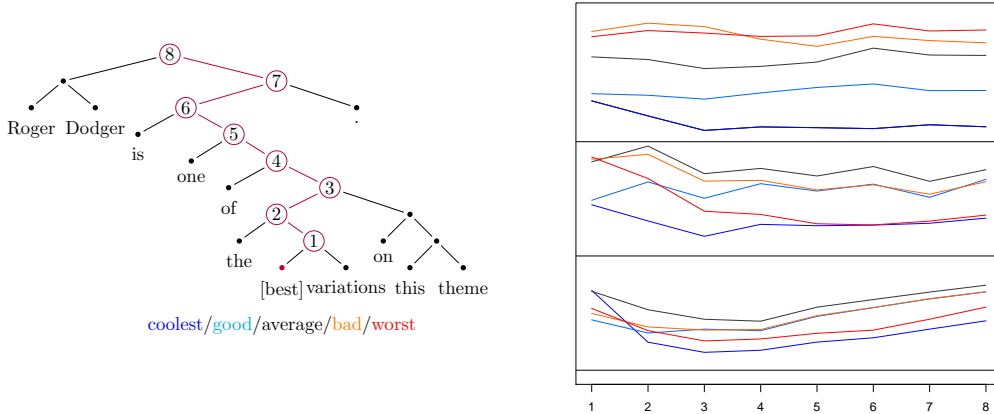

Figure 3: An example sentence with its parse tree (left) and the response measure of every layer (right) in a three-layered deep recursive net. We change the word *"best"* in the input to one of the words *"coolest"*, *"good"*, *"average"*, *"bad"*, *"worst"* (denoted by blue, light blue, black, orange and red, respectively) and measure the change of hidden layer representations in one-norm for every node in the path.

| charming results | | |
|---|---|---|
| 1  charming , | interesting results | charming chemistry |
| 2  charming and | riveting performances | perfect ingredients |
| 3  appealingly manic and energetic | gripping performances | brilliantly played |
| 4  refreshingly adult take on adultery | joyous documentary | perfect medium |
| 5  unpretentious , sociologically pointed | an amazing slapstick instrument | engaging film |
| not great | | |
| 1  as great | nothing good | not very informative |
| 2  a great | not compelling | not really funny |
| 3  is great | only good | not quite satisfying |
| 4  Is n't it great | too great | thrashy fun |
| 5  be great | completely numbing experience | fake fun |

Table 2: Example shortest phrases and their nearest neighbors across three layers.

In Figure 3, we give an example sentence, *"Roger Dodger is one of the best variations on this theme"* with its parse tree. We change the word *"best"* into the set of words *"coolest"*, *"good"*, *"average"*, *"bad"*, *"worst"*, and measure the response of this change along the path that connects the leaf to the root (labeled from 1 to 8). Note that all other nodes have the same representations, since a node is completely determined by its subtree. For each node, the response is measured as the change of its hidden representation in one-norm, for each of the three layers in the network, with respect to the hidden representations using the original word (*"best"*).

In the first layer (bottom) we observe a shared trend change as we go up in the tree. Note that *"good"* and *"bad"* are almost on top of each other, which suggests that there is not necessarily enough information captured in the first layer yet to make the correct sentiment decision. In the second layer (middle) an interesting phenomenon occurs: Paths with *"coolest"* and *"good"* start close together, as well as *"worst"* and *"bad"*. However, as we move up in the tree, paths with *"worst"* and *"coolest"* come closer together as well as the paths with *"good"* and *"bad"*. This suggests that the second layer remembers the *intensity* of the sentiment, rather than *direction*. The third layer (top) is the most consistent one as we traverse upward the tree, and correct sentiment decisions persist across the path.

**Nearest Neighbor Phrases.** In order to evaulate the different notions of similarity in the meaning space captured by multiple layers, we look at nearest neighbors of short phrases. For a three layer deep recursive neural network we compute hidden representations for all phrases in our data. Then, for a given phrase, we find its nearest neighbor phrases across each layer, with the one-norm distance measure. Two examples are given in Table 2.

For the first layer, we observe that similarity is dominated by one of the words that is composed, i.e. *"charming"* for the phrase *"charming results"* (and *"appealing"*, *"refreshing"* for some neighbors), and *"great"* for the phrase *"not great"*. This effect is so strong that it even discards the negation for the second case, *"as great"* and *"is great"* are considered similar to *"not great"*.

In the second layer, we observe a more diverse set of phrases semantically. On the other hand, this layer seems to be taking syntactic similarity more into account: in the first example, the nearest neighbors of *"charming results"* are comprised of adjective-noun combinations that also exhibit some similarity in meaning (e.g. *"interesting results"*, *"riveting performances"*). The account is similar for *"not great"*: its nearest neighbors are adverb-adjective combinations in which the adjectives exhibit some semantic overlap (e.g. *"good"*, *"compelling"*). Sentiment is still not properly captured in this layer, however, as seen with the neighbor *"too great"* for the phrase *"not great"*.

In the third and final layer, we see a higher level of semantic similarity, in the sense that phrases are mostly related to one another in terms of sentiment. Note that since this is a supervised task on sentiment detection, it is sufficient for the network to capture only the sentiment (and how it is composed in context) in the last layer. Therefore, it should be expected to observe an even more diverse set of neighbors with only a sentiment connection.

## 4   Conclusion

In this work we propose the deep recursive neural network, which is constructed by stacking multiple recursive layers on top of each other. We apply this architecture to the task of fine-grained sentiment classification using binary parse trees as the structure. We empirically evaluated our models against shallow recursive nets. Additionally, we compared with previous work on the task, including a multiplicative RNN and the more recent Paragraph Vectors method. Our experiments show that deep models outperform their shallow counterparts of the same size. Furthermore, deep RNN outperforms the baselines, achieving state-of-the-art performance on the task.

We further investigate our models qualitatively by performing input perturbation, and examining nearest neighboring phrases of given examples. These results suggest that adding depth to a recursive net is different from adding width. Each layer captures a different aspect of compositionality. Phrase representations focus on different aspects of meaning at each layer, as seen by nearest neighbor phrase examples.

Since our task was supervised, learned representations seemed to be focused on sentiment, as in previous work. An important future direction might be an application of the deep RNN to a broader, more general task, even an unsupervised one (e.g. as in [9]). This might provide better insights on the operation of different layers and their contribution, with a more general notion of composition. The effects of fine-tuning word vectors on the performance of deep RNN is also open to investigation.

**Acknowledgments**

This work was supported in part by NSF grant IIS-1314778 and DARPA DEFT FA8750-13-2-0015. The views and conclusions contained herein are those of the authors and should not be interpreted as necessarily representing the official policies or endorsements, either expressed or implied, of NSF, DARPA or the U.S. Government.

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
