[Reviews · NeurIPS 2014]

Submitted by Assigned_Reviewer_29

Paper For paper 1180: Deep Recursive Neural Networks for Compositionality in Language

This paper introduces a new architecture — deep recursive neural network (deep RNN) which
is constructed by stacking multiple recursive layers. The authors evaluate the proposed
model on the task of fine-grained sentiment classification.

Clarity
- In general, this paper is well written and pleasant to read.

Quality
- The paper seems technically sound.
- The major idea is nice, i.e. the deep recursive neural network, which is constructed by stacking multiple layers of individual recursive nets.
- The authors provided reasonable amounts of experimental results to evaluate various aspects of the method

Originality
- The proposed idea is simple and reasonable
- It might be better to add more discussions about
+ why intuitively deeper network is good for the target task?
+ why choose the selected network architecture ?

Significance
- The paper has presented interesting idea, clear background description and reasonable experimental supports.
Summary: - The paper has presented interesting idea, clear background description and reasonable experimental supports.

Submitted by Assigned_Reviewer_31

This exciting paper introduces a new deep recursive neural architecture that obtains state of the art performance on a hard sentiment classificationd dataset.

The results from their new architecture are impressive and the exposition is very clear!

The changes they propose to previous RNN models are clever, well motivated and experimentally demonstrated to work well:
- untying leaf nodes and nonterminal nodes
- using rectified linear units in the recursive setting
- using large unsupervised word vectors but smaller hidden nonterminal nodes
- using a deep architecture which is not simply replacing a single neural network RNN layer with a deep layer but introducing connections between the layers and tree nodes. -- it would have been an interesting comparison if the outputs of the last hidden layer at each node was the only input to the next parent, i.e. to drop the connection from matrix V?

Table 1 (a) results should have been on the dev set, not the final test set. you're tuning ont he final test set here!

typos:
- comprise a class of architecture
Summary: This exciting paper introduces a new deep recursive neural architecture that obtains state of the art performance on a hard sentiment classificationd dataset.

The results from their new architecture are impressive and the exposition is very clear!

Submitted by Assigned_Reviewer_32

This paper proposes the use of 'deep' recursive networks by adding depth across space, which is motivated by the success of doing the same thing with recurrent networks. Each layer has its own parameters that are shared across different heights within that layer. The authors also propose the use of dropout and ReLUs, while fixing the word representations to pre-trained embeddings. Experimentation is performed on the sentiment treebank of Socher et al.

The proposed method is novel although essentially a generalization of the same approach used with recurrent nets. The authors achieve very strong performance on the sentiment treebank dataset. The performance gains obtained with dropout and ReLUs are interesting in itself. That being said, some of the model and experimental design choices are questionable (or not well justified) and I believe some additional experimentation is necessary before I can wholeheartedly recommend acceptance.

The paper itself is clearly written for the most part and sufficient detail is given for the reader to reproduce the results on their own.

Detailed comments / questions:

- line 138: What is W^eta and W^xh? These haven't been defined. Are these just to refer to either W^{eta}_L and W^{eta}_R? It would be clearer to have two equations: one for the inputs and another for the hidden layers. It would also help to summarize the parameter space i.e. {W_L, W_R, U, b, c}. Including the dimensionalities of each of your matricies when you define them would also be helpful.

- lines 148-154: This seems very speculative. Is this based off of your own empirical results? If so, you should mention this. I'm not convinced that this is actually an issue.

- baseline: The authors should include the results of "A convolutional neural network for modelling sentences" (Kalchbrenner et al) from ACL this year.

- How come you didn't fine-tune the word vectors? Finetuning them on a sentiment task allows the words embeddings to become reflective of their sentiment. It seems strange to have phrases and sentence vectors that are discriminative of sentiment but where individual word vectors are not. (I realize this is OK with a deep RNN since you could just forward-pass the word vectors across layers). Is it possible that the improvements you're getting with deep RNNs happening because you are not fine-tuning the words? Conceivably, a single layer RNN with fine-tuned embeddings should do better. I think this needs to be controlled for.

- line 251 (shared dropout units): What was the motivation for this choice?

- lines 255-256 (stability): Alternatively, you could try constraining the norms of the weights or truncating the gradients, as is sometimes done with recurrent nets.

- binary classification: This should be done separately. The issue is that at training time, you are also including the neutral class examples which deviates from the experiment protocol. Consequently, these results are not directly comparable to the existing approaches.

- The analogy with deep recurrent nets is reasonable when you are making predictions at each node in the tree, as is the case on the treebank experiments. It would have been useful to see whether or not there is an advantage to using deep RNNs when only a single, root label exists. Would you still expect to see an improvement over a single layer RNN?
Summary: The use of deep RNNs seems like a promising approach but I would like to see some additional experiments. I would recommend that the authors redo the binary classification experiment, as per the proper protocol, study the effect of fine-tuning the word embeddings and finally include 1-2 additional datasets that only use a global label.

These additional experiments would result in a much stronger paper as well as stronger evidence for the usefulness of depth across space.
Author Feedback
Author rebuttal: For brevity, we are responding primarily to Reviewer 3.

line 138: Yes, small dot as a subscript refers to one of L or R. We will make the notation more explicit, and add a summary of parameters as well as the matrix dimensions in the final version, as you suggest.

lines 148-154: This is based purely on the definitions of the ReLU activation and the word vectors that we use. Specifically, word vectors belong to a dense space, whereas the ReLU-activated hidden layers belong to a sparse space (we will include citations for this, e.g. Glorot et al, 2011); thus, the two manifolds are different from one another in this respect. We believed that this justifies the use of unshared weights. Of course in principle, a single layer can handle both manifolds, by making a distinction between the two manifolds and acting (potentially differently) on the input. This, however, might require more representational power from the network.

baseline: ACL was after the deadline for NIPS, hence we were not able to cite this. We will describe it in the related work section of to the final version and include it in the evaluation. Because the same training/test partitions are employed, the Kalchbrenner et al. results can be directly compared to ours; our deep RNNs outperform theirs by 1.3 absolute difference in fine grained accuracy.

fine-tuning: Fine-tuning word vectors adds another layer of complexity to the architecture. In particular, it adds the learning rate of the word vectors as another hyperparameter to tune. Based on our own experiments, we found that the simple solution of using the same learning rate as the rest of the network results in severe overfitting. To keep things simpler and mainly to focus on the architectural differences between shallow and deep RNNs, and how deep RNNs represent input, we chose not to fine-tune.

line 251 (shared dropouts): Experts among deep-learning community strongly suggest sharing the dropped units, when sharing connections, such as in recurrent neural nets or convolutional nets. This is also in line with the 'ensemble' interpretation of dropouts (Hinton et al, 2012).

lines 255-256 (stability): Yes, we definitely acknowledge that there are alternatives for maintaining stability. We simply selected one of the common solutions known to work, since this was not the main focus of the paper.

binary classification: Our decision to include the neutral class was based in part on direct correspondence with Richard Socher. In particular, in the original deep recurrent networks paper (Socher et al. 2013) did not explicitly state how they performed the binary experiments. So we asked the authors whether we should (a) redo our experiments to obtain binary (positive-negative) scores (in this case throwing out neutral cases is not trivial because non-neutral roots can include neutral subtrees) or (b) decode the multiclass results into binary. Socher responded saying that either method is fine, so we picked the simplest option that did not require extra training. Also this potentially negatively affects our results, because we are evaluating the model on a slightly different task than it is trained on. So some power of the network that distinguishes between two classes of same polarity, or a neutral from a non-neutral, is wasted when evaluating on binary classification.

Single output per tree: Yes, experiments to evaluate the networks when there is only a single label at the root would be quite interesting. We had been planning it as part of our future work since the experimental setting is quite different; our goal was to show the promise of deep RNNs for semantic compositionality in natural language for which the multi-class sentiment prediction data set (that we employed) is known to be difficult. In addition, Socher et al. have successfully applied recursive nets for tasks with a single error source; therefore, at least in terms of representational power, deep recursive nets should be able to handle them.

Reviewer 2: Yes, we can also include "model selection" experiments that are done on the development set in conjunction with the results in Table 1(a).